# Spatial Prediction of Wildfire Susceptibility Using Hybrid Machine Learning Models Based on Support Vector Regression in Sydney, Australia

Arip Syaripudin Nur [1], Yong Je Kim [2], Joon Ho Lee [1] and Chang-Wook Lee [1,3,*]

[1] Division of Science Education, Kangwon National University, Chuncheon-si 24341, Republic of Korea
[2] Department of Civil and Environmental Engineering, Lamar University, 4400 MLK Blvd., Beaumont, TX 77710, USA
[3] Department of Smart Regional Innovation, Kangwon National University, Chuncheon-si 24341, Republic of Korea
* Correspondence: cwlee@kangwon.ac.kr

**Abstract:** Australia has suffered devastating wildfires recently, and is predisposed to them due to several factors, including topography, meteorology, vegetation, and ignition sources. This study utilized a geographic information system (GIS) technique to analyze and understand the factors that regulate the spatial distribution of wildfire incidents and machine learning to predict wildfire susceptibility in Sydney. Wildfire inventory data were constructed by combining the fire perimeter through field surveys and fire occurrence data gathered from the visible infrared imaging radiometer suite (VIIRS)-Suomi thermal anomalies product between 2011 and 2020 for the Sydney area. Sixteen wildfire-related factors were acquired to assess the potential of machine learning based on support vector regression (SVR) and various metaheuristic approaches (GWO and PSO) for wildfire susceptibility mapping in Sydney. In addition, the 2019–2020 "Black Summer" fire acted as a validation dataset to assess the predictive capability of the developed model. Furthermore, the information gain ratio (IGR) method showed that driving factors such as land use, forest type, and slope degree have a large impact on wildfire susceptibility in the study area, and the frequency ratio (FR) method represented how the factors influence wildfire occurrence. Model evaluation based on area under the curve (AUC) and root average square error (RMSE) were used, and the outputs showed that the hybrid-based SVR-PSO (AUC = 0.882, RMSE = 0.006) model performed better than the standalone SVR (AUC = 0.837, RMSE = 0.097) and SVR-GWO (AUC = 0.873, RMSE = 0.080) models. Thus, optimizing SVR with metaheuristics improved the accuracy of wildfire susceptibility modeling in the study area. The proposed framework can be an alternative to the modeling approach and can be adapted for any research related to the susceptibility of different disturbances.

**Keywords:** wildfire; Sydney; VIIR; support vector regression; susceptibility map

## 1. Introduction

Wildfires are a natural, complex, and important part of the Australian environment [1]. Wildfires can be triggered by natural causes, such as lightning strikes, or by humans (intentionally or unintentionally), and weather and fuel conditions also play a role in their creation [2,3]. Small branches, leaf litter, twigs and bark, shrubs, and grasses can provide material to burn for wildfires. How much the material that is available to burn, its type, and how moist or dry it is will influence wildfire conditions. Windy, hot, and dry weather can contribute to the fire hazard. Throughout the year, some regions in Australia are vulnerable to wildfires. In northern Australia, the dry season is the peak wildfire period, which generally occurs in spring and winter. As for southern Australia, the peak of wildfire season is in the fall and summer. Although these are the traditional peaks of the wildfire season, local conditions may trigger wildfire hazard activity at any time [4].

Wildfires cause significant damage to the environment, ecology, economy, and threaten human life and assets [5]. During the "Black Saturday" wildfire, 173 lives were lost, and 4500 km$^2$ of area was burned in Australia in 2009 [6]. In 2013, 248 buildings were destroyed by wildfires across New South Wales (NSW) [6]. The "Black Summer" fire season in the summer of 2019/2020 saw the most destructive wildfire, which burned about 34 million hectares, over 3000 houses, and caused economic losses of over AU$ 100 billion [7–9]. Sydney, the city with the highest population in Australia, is located in the southeast of the country, where these fires are the most widely spread [10]. Radiant heat and smoke emission are other impacts of wildfires [11]. Radiant heat can be felt more than 100 m away from a large wildfire and can potentially damage objects, such as vehicles and structures. Toxic fumes and thick smoke produced from wildfires can interfere with vision and affect the air quality and make breathing difficult [12]. Burning embers can fly and spread several kilometers from the site of a large wildfire, causing small hotspots to break out. Due to the unpredictable and fast-spreading nature of wildfires, people are advised to evacuate from their homes as quickly as possible to ensure their safety. Despite efforts to increase the allocation of resources and strategic planning for fire suppression in recent years, extensive human activities and climate change have increasingly been determined as important factors that drive the trend of more wildfires, resulting in larger burned areas globally [3,13,14]. It is estimated that the severity, burnt area, and number of wildfires will escalate in the future due to climate change [15,16]. Prolonged dry seasons and high temperatures might result in unexpected wildfire activities [15]. According to the state temperature dataset in Australia, the warmest year on record was 2019, with the annual national average temperature 1.52 °C above average. The driest year on record in Australia was also in 2019, with notable heatwaves in January and December. However, a recent study in California showed that human activities might be a significant factor rather than natural circumstances with regard to wildfire ignition [3]. Therefore, it is necessary to model wildfires to identify areas with a high possibility of wildfire events for establishing better wildfire risk management to decrease the negative impacts of wildfires on humans and the environment [5,17–19].

The improvement of remote sensing technologies has advanced wildfire monitoring and management across the globe. With the improved 375 m spatial resolution data that complements the MODIS fire observation, the visible infrared imaging radiometer suite (VIIRS) data provides better fire sensitivity over relatively small areas and has improved nighttime performance [20]. Fire susceptibility modeling uses various techniques that usually utilize the geographic information system (GIS) method and remote sensing (RS) data. Well known methods with wide applications include logistic regression [21], the analytical hierarchy process (AHP) [22], fuzzy systems [23,24], weight of evidence (WOE) [25], artificial neural networks (ANN) [13], the evidential belief function [26], and decision trees [27]. However, the selection of parameters has a significant influence on the learning phase, determining the results of predictions, and has the potential to lead to various problems such as overfitting or underfitting. The use of a precise parameter estimation approach is an important part of the success of modeling efforts, especially regarding machine learning models. The disadvantage of conventional parameter computation approaches is the utilization of a complicated seeking process to identify the optimal values of the parameter [28]. For instance, the gradient descent learning [29] and evolutionary learning [30,31] methods often used for adjusting the SVR algorithm require long processing times and decrease the effectiveness of the model. The metaheuristic optimization approach can be utilized to overcome the ineffective slow parameter estimation process of the models [32]. Therefore, there is no need for trial and error tests by using the automatic computation through metaheuristic algorithms to determine optimal parameters [33]. The GWO and PSO algorithms have proven to be useful for various susceptibility mapping, such as for landslides [28,34], groundwater [35] and floods [36], and therefore have been evaluated for use in SVR model tests.

In previous studies, Sulova and Arsanjani (2020) [19] applied three different machine learning algorithms at the continental level using random forest, naïve Bayes and regression tree models, and Hosseini and Lim (2022) [5] mapped at the state level using eight methods. This study aims to create wildfire susceptibility prediction at the regional level using VIIRS Suomi data and hybrid models based on the metaheuristic optimization of powerful machine learning models. The standalone SVR model and its metaheuristic optimized versions, SVR-GWO and SVR-PSO, were used to create wildfire susceptibility maps in Sydney and surrounding areas in southeastern Australia. As a matter of convenience, Table 1 represents the nomenclature of this paper.

**Table 1.** This table represents the nomenclature of this paper.

| Nomenclature | | | |
|---|---|---|---|
| MODIS | moderate resolution imaging spectroradiometer | $H$ | training dataset |
| VIIRS | visible infrared imaging radiometer suite | $RF$ | related factors |
| GIS | geographic information system | FR | frequency ratio |
| RS | remote sensing | $n$ | number of data points or samples |
| AHP | analytical hierarchy process | $y_i$ | output values |
| ANN | artificial neural network | $x_i$ | input data |
| WOE | weight of evidence | $w^T$ | the transpose value of weight factor |
| SVR | support vector regression | $b$ | bias vectors |
| GWO | grey wolf optimization | $\varphi(x)$ | nonlinear function |
| PSO | particle swarm optimization | $C$ | penalty factor |
| FIRMS | fire information for resource management system | $\xi_i$ | loose variables or distance between boundary |
| RMSE | root mean square error | $\xi_i{}^*$ | targets |
| ROC | the receiver operating characteristic | $\varepsilon$ | insensitive loss function |
| AUC | area under ROC curve | $\alpha_i$ | Lagrange multipliers |
| NSW | New South Wales | $k(x,x_i)$ | the kernel function |
| Cfa | humid subtropical climate | $\overrightarrow{X_i}$ | particle location |
| S-NPP | Suomi-national polar-orbiting partnership | $\overrightarrow{V_i}$ | particle velocity |
| GPS | global positioning system | $t$ | iteration number |
| NDVI | normalized difference vegetation index | $w$ | inertial weight |
| DEM | digital elevation model | $Pi$ | the best position of particle $i$ |
| ABARES | Australian bureau of agricultural and resource economics and sciences | $G$ | the fittest position of the entire swarm |
| PDSI | Palmer drought severity index | $c_1$ | cognitive acceleration constant |
| ACLUMP | Australian collaborative land use management program | $c_2$ | social acceleration coefficient |
| IGR | information gain ratio | $r$ | random coefficients range from 0 to 1 |
| TOL | tolerance | $p$ | predicted value |
| VIF | variance inflation factor | $o$ | actual value |

## 2. Materials and Methods

The methodology of this study is outlined in a graphical illustration in Figure 1. The first step was producing a wildfire database using the FIRMS dataset from the VIIRS-Suomi satellite. The 2011–2018 wildfire location data were then divided into training (70%) and testing (30%) datasets using a random function. Afterward, a spatial database consisting of wildfire-related factors was constructed and assessed using spatial relationship analysis using the training set and layers of related factors. Using GWO and PSO metaheuristic optimization algorithms, the optimal hyperparameters of SVR were then identified, and the susceptibility models were created. At last, the resulting maps were validated utilizing the 2019–2020 Black Summer Fire datasets using RMSE and AUC analysis.

### 2.1. Study Area

The study area is Sydney and the surrounding county in the eastern part of New South Wales, Australia, as shown in Figure 2. Sydney is home to 5,259,764 people, or approximately 20.20% of the Australian population [37]. The New South Wales (NSW) National Park and Wildlife Service contains 50 parks (national, conservation, reserve, etc.) with Wollemi National Park as the largest announced wilderness area in NSW [38]. For simplicity, the study area is denoted as Sydney. Sydney has an area of 48,121 km$^2$, and its elevation ranges from 0 to 1356 m. About 72% of the area has slope degrees from 0 (flat) to 150, with the highest slope inclination of 81 degrees. The urban areas are mostly

located in low-lying areas near the coast. The climate of Sydney is humid subtropical (Koppen: cfa) [39], with mild winters and warm summers. The land cover types of the study area include conservation and natural environment areas (45.31%), production areas from relatively natural environments (24.85%), dryland agriculture (17.02%), wetland agriculture (0.64%), and urban areas (10.37%). The forestry area of Sydney is dominated by eucalyptus (52%), followed by other native forests (2.28%), and Casuarina (1.91%).

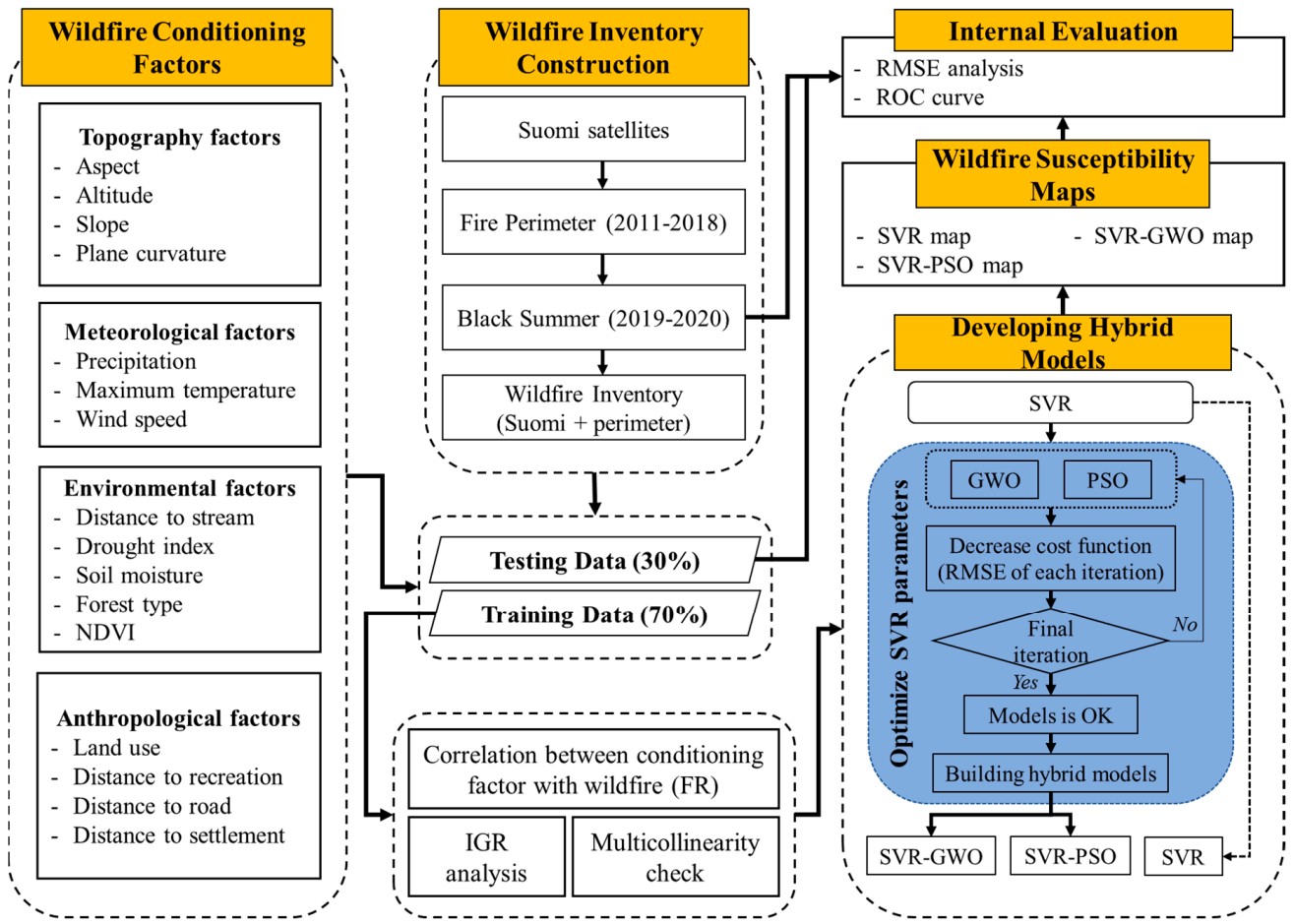

**Figure 1.** Flowchart of the overall methodology.

### 2.2. Historical Fire Location

Creating an inventory map is mandatory before the construction of wildfire suscepti­bility maps [40]. The wildfire inventory in Sydney was collected from the Fire Information for Resource Management System (FIRMS) acquired from the VIIRS instrument attached to the Suomi-National Polar-orbiting Partnership (S-NPP) satellite from 2011 to 2018. Each pixel of VIIRS active thermal/fire hotspot location depicts the center of a 375 m resolution. These data files include the latitude, longitude, acquisition time and date, and confidence level (low, normal, and high). Only high confidence level data were used as true fire hotspots for the wildfire inventory database to provide more certain and accurate fire location information. From 2011–2018, 26,258 samples were acquired. The data were then compared with fire perimeters generated by the Australian government and sourced fire datasets from the NSW Rural Fire Service and Forestry Corporation of NSW. The dataset was derived from hand digitizing fire scars from aerial photography and GPS coordinates. Finally, a combination of Suomi data and fire perimeters provided 16,462 samples for the fire inventory.

In machine learning processing, the construction of a wildfire susceptibility model was conducted as a binary classification, which required data from fire and without fire areas.

An equal number of nonwildfire location data (16,462 points) were picked using the random point function by identifying the region outside previous wildfire history and having very low possibility areas determined using a frequency ratio approach. This approach was an effective strategy to help the interpretation of the area, and provides a more precise wildfire inventory. The wildfire and nonwildfire data were split into training (70%) and testing (30%) datasets because this ratio was found in many studies in which this integration was appropriate to predict wildfire susceptibility maps [5,41,42]. The 70% training datasets for wildfire-occurrence and nonwildfire-incident locations were then combined to produce wildfire susceptibility maps, and 30% of the test data from the two datasets were combined for validation of the model's performance.

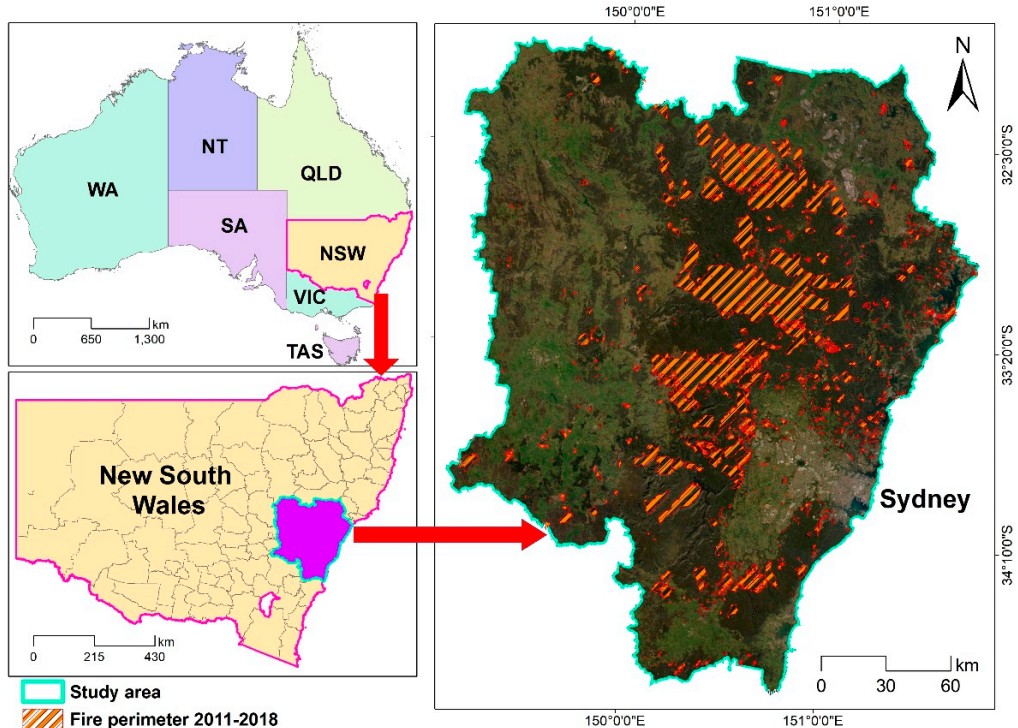

**Figure 2.** The location of the study area (cyan line) in Sydney, Australia, and the polygons of location of burned areas from 2011 to 2018 in Sydney and the surrounding area.

## 2.3. Additional Fire Data for Validation

To evaluate the predictive capability, we considered using 2019–2020 fire data during the Black Summer wildfire. The validation data constituted a total of 32 wildfires that took place from October 2019 to May 2020. The smallest area of wildfire was 109 hectares, namely the Spring Gully Fire, which occurred on 9 January 2020. The largest wildfire was the Gospers Mountain fire, which burned 479.513 hectares from 25 October 2019 until 9 February 2020. The wildfire data were also collected from VIIRS-Suomi data integrated with the fire perimeter from the Australian government. Therefore, we collected 61,838 samples that act as a validation dataset for wildfire susceptibility models.

## 2.4. Wildfire-Related Factors

The selection of independent variables, which are also called as predisposing, predictors, conditioning, or driving variables, is essential in susceptibility modeling. Table 2 shows 16 variables related to wildfire susceptibility that were selected according to data availability and prior studies in Australia [5,19]. The variables were categorized into four categories including the topographical, environmental, meteorological, and anthropological. The topographical-related factors included altitude, aspect, plan curvature, and slope; the meteorological-related factors include windspeed, maximum temperature, and precipitation; the environmental-related factors include soil moisture, distance to rivers,

drought index, forest type, and the normalized difference vegetation index (NDVI); and the anthropological-related factors include distance to recreational areas, land use, distance to roads, and distance to human settlements. Figures 3–6 show the 16 factors utilized for the wildfire susceptibility assessment. All factors were arranged into a spatial database and were resampled to 30 m spatial resolution. Using the quantile method, numeric or continuous data were reclassified into five classes to determine and analyze the effect of wildfires in each class.

**Table 2.** Information on wildfire driving factors.

| Category | Variable Name | Resolution | Source of Data | Ref. |
|---|---|---|---|---|
| Topographical | Altitude | 30 m | Copernicus DEM | [43] |
| | Aspect | | | [44] |
| | Plan curvature | | | [13] |
| | Slope | | | [45] |
| Meteorological | Precipitation | 4 km | Terra climate | [46] |
| | Maximum temperature | | | [18] |
| | Windspeed | 50 m | Global Wind Atlas | [42] |
| Environmental | Dist. to rivers | 50 m | ABARES | [47] |
| | Drought index | 4 km | Terra climate | [48] |
| | Soil moisture | | | [19] |
| | Forest type | 100 m | ABARES | [49] |
| | NDVI | 375 m | MODIS | [50] |
| Anthropological | Land use | 50 m | ABARES | [51] |
| | Dist. to recreational areas | | | [52] |
| | Dist. to roads | | | [53] |
| | Dist. to human settlements | | | [21] |

### 2.4.1. Topographical

Topography-related factors have an essential effect on wildfire occurrence, distribution, severity of vegetation, human accessibility, and local climate [17]. Topographic factors (Figure 3) include altitude, aspect, plan curvature, and slope, which are derived from the Copernicus DEM (30 meter spatial resolution). Altitude influences the severity and spread of a wildfire and is related with plant composition and distribution and conditions of the local climate [54]. Escalating the slope degree can intensify the speed of fire distribution. Fires can distribute less quickly in less steep zones and faster in steep zones. The aspect indicates the direction the surface is facing and influences how much sunlight is received [25].

### 2.4.2. Meteorological

Meteorological-related factors (Figure 4) include the average values of maximum temperature, precipitation, and windspeed data. Meteorological factors regulate the cycle life of vegetation, which provides drying leaves for ignition, fuel production, or spreading wildfires [43]. Precipitation and maximum temperature data were gathered through the Terra Climate 2011–2018 dataset (4 km spatial resolution). This monthly data uses the interpolated time-varying anomalies from CRU Ts4.0/JRA55 to produce a dataset that covers a larger temporal record [19]. Precipitation influences the plants pattern, and moisture levels affect the rate of fire spread. The higher the temperature, the higher the likelihood that a fire will ignite or keep on burning. The reason for this is that at high temperatures, the fuel becomes closer to its ignition point, and preheated fuel burns faster. Windspeed data was collected from the Global Wind Atlas (50 m spatial resolution). Strong winds blow the flames and cause a fire to spread quickly across the landscape. Strong winds can also take carry fire embers a long way, which can trigger spot fires many kilometers ahead of the main fire front. The raster was derived from data acquired from 2011 to 2018 by applying the average statistical tools.

### 2.4.3. Environmental

Environmental-related factors (Figure 5) include distance to rivers, the drought index, soil moisture level, forest type, and NDVI data. River- and forest-type data were acquired from the Australian Bureau of Agricultural and Resource Economics and Sciences (ABARES) [55]. The distance to rivers is related to the condition of the forest with the river that serves as a water source. The soil moisture level and drought index data were gathered through the Terra Climate data from 2011 to 2018. The drought index data estimates landscape drought and surface water balance. This data was based on the Palmer drought severity index (PDSI) approach [56]. Soil moisture affects the dryness degree and water balance of fuels and influences the dead plants lying on the surface [14]. The forest-type data is essential for the assessment of wildfires in Sydney since it serves as information about the plant distribution and characteristics of the study area [57]. This information helped in identifying which type of vegetation had to be maintained and prioritized since flammability may vary according to the type of vegetation. The NDVI data was gathered from the MODIS/Terra satellite (375 m). The NDVI data were calculated using the median function from 2011 to 2018 to avoid bias caused by the loss of greenness after wildfires. The NDVI represents the condition, health, and moisture of vegetation [21]. A decrease in the NDVI shows that dry vegetation influences water stress and increases the likelihood of wildfire.

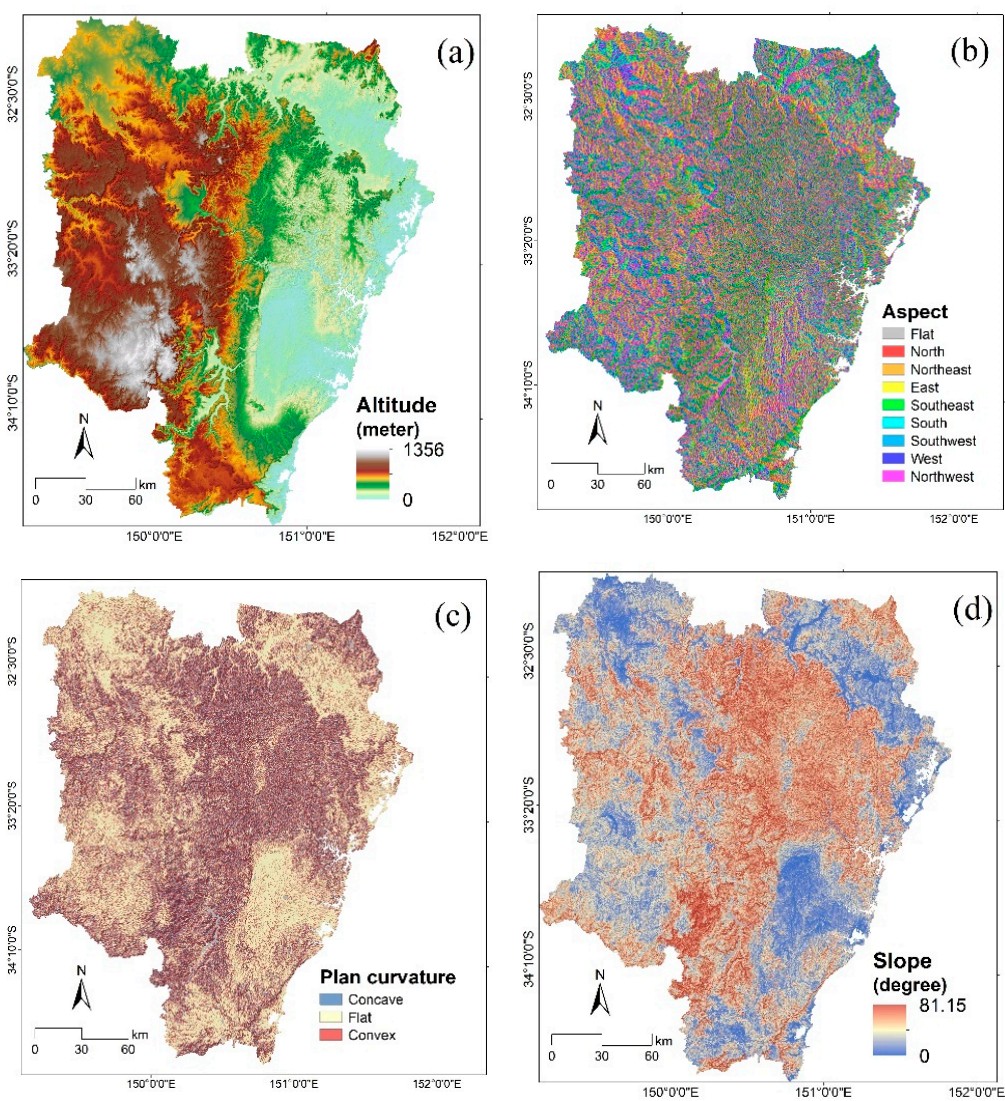

**Figure 3.** Topographical-related factors: (**a**) altitude, (**b**) aspect, (**c**) plan curvature, and (**d**) slope.

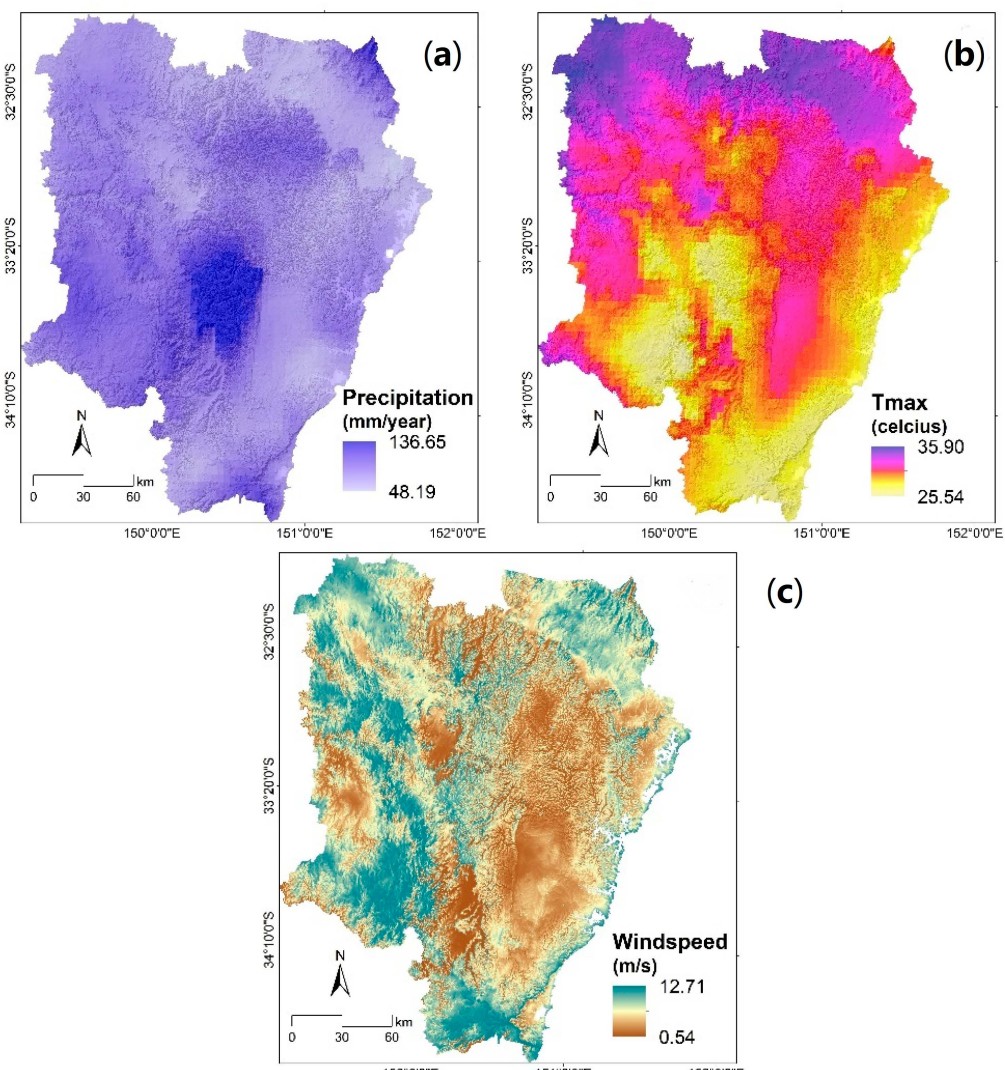

**Figure 4.** Meteorological-related factors: (**a**) precipitation, (**b**) maximum temperature, and (**c**) windspeed.

### 2.4.4. Anthropological

Anthropological-related factors (Figure 6) include land use, distance to recreational areas, roads, and settlements, which were collected from ABARES (50 m) [58]. Updated in December 2020, this product consists of national compilation data from the Australian Collaborative Land Use Management Program (ACLUMP). Land use depicts the landscape pattern, composition, and characteristics of the study area. These features may influence the triggering and distribution of fires. The distance to recreational areas, roads, and human settlements quantify the accessibility to forest areas and wildfire areas and, in many cases, human activity is responsible for the triggering of wildfires.

### *2.5. Spatial Correlation Analysis*

Feature selection approaches can be used to determine and eliminate noisy, irrelevant, or redundant data that may deteriorate model accuracy. Here, multicollinearity, Pearson correlation, and information gain ratio (IGR) approaches were applied to assess the spatial correlation between several wildfire-driving factors. Multicollinearity examination can be utilized to identify the existence of correlated wildfire-driving factors.

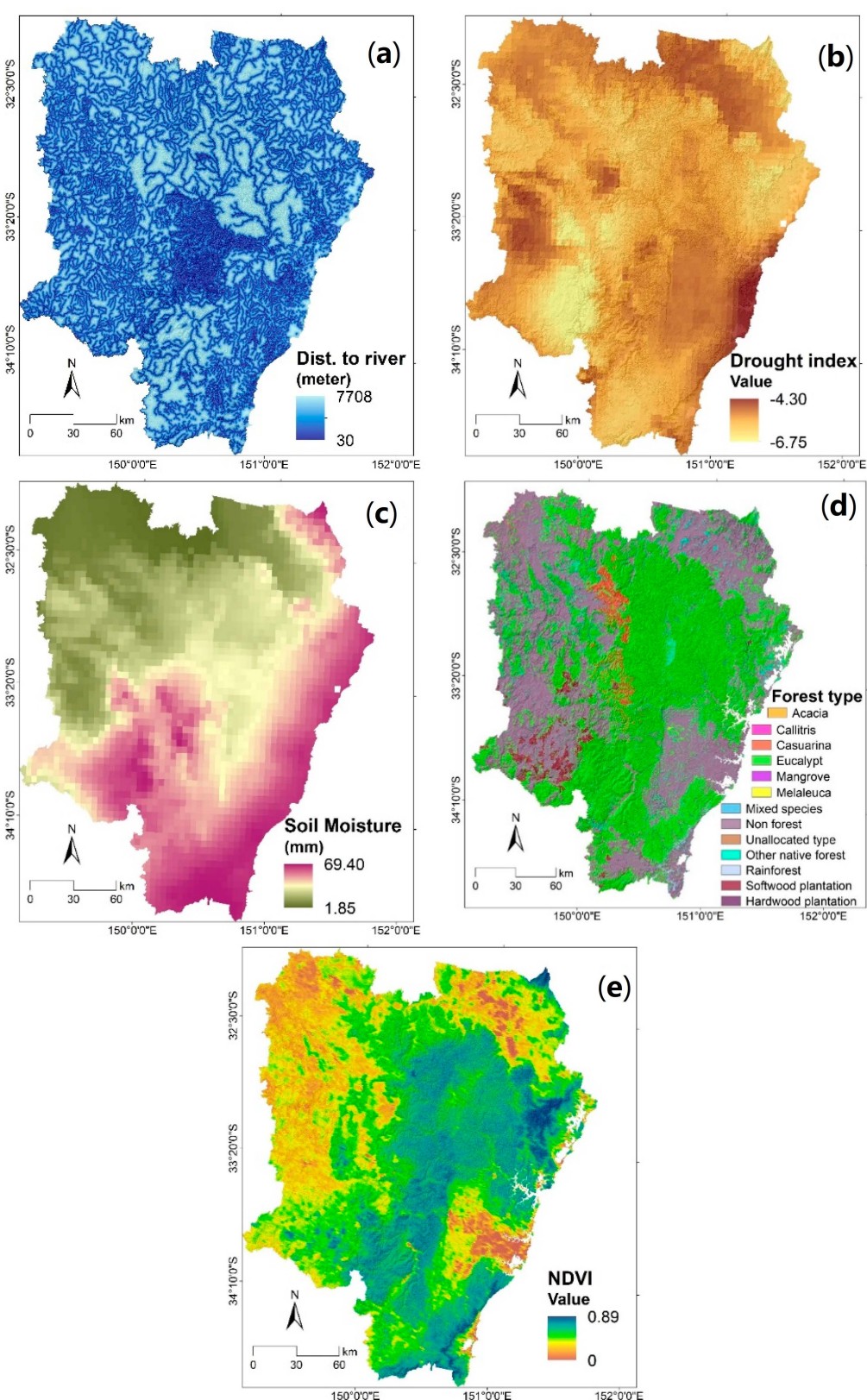

**Figure 5.** Environmental-related factors: (**a**) distance to rivers, (**b**) drought index, (**c**) soil moisture precipitation, (**d**) forest type, and (**e**) NDVI.

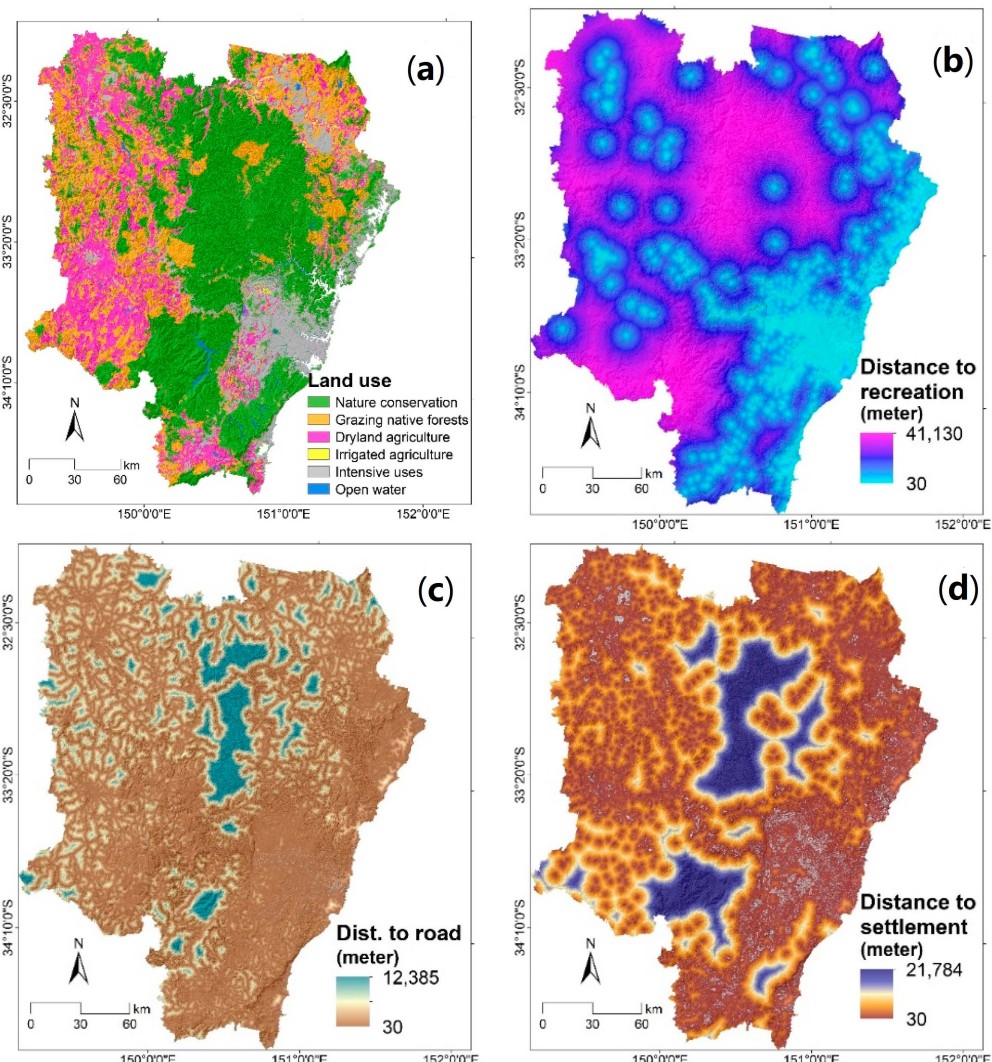

**Figure 6.** Anthropological-related factors: (**a**) land use, (**b**) distance to recreational areas, (**c**) distance to roads, and (**d**) distances to settlements.

Multicollinearity may occur among some factors if they are not accurately selected, and the factors should be removed [41]. The methods are tolerance (TOL) and the variance inflation factor (VIF), which are two common indexes for indicating multicollinearity. If a TOL score is less than 0.1, or a VIF score is greater than 10, this suggests the existence of multicollinearity [59]. A Pearson's correlation coefficient calculation was conducted to determine the strength and direction of correlation among wildfire-driving variables to support the multicollinearity test. The results scores range between −1 and +1. Scores near −1 imply a strong negative correlation, while scores near +1 denote that there is a strong positive relationship between two driving variables. Scores equal to 0 indicate no correlation between the two factors. Prior studies have considered the scores −0.7 and 0.7 to be the critical score, implying that scores above these critical scores may lead to a multicollinearity problem [17,60,61].

Furthermore, the IGR analysis has also been utilized by other studies [17,62] to choose the appropriate driving factors for wildfire susceptibility maps. The concept of IGR is an entropy-based feature selection approach, and it is employed to provide information from

the feature items and to identify the total entropy reduction of the database [63]. A high score on the IGR represents a better predictive ability of a driving factor [64].

$$IGR\ (H, RF) = \frac{Entropy(H) - Entropy(H,\ RF)}{Split\ Entropy(H)} \tag{1}$$

where *H* represents the training dataset, *RF* represents the related factors, and *Entropy*(*H*, *RF*) is the information gathered by dividing the training data (*H*) and related factors (*RF*).

The frequency ratio (FR) method was employed to calculate and analyze the spatial correlations between the wildfire locations (11,523 points) and classes of each wildfire-driving factor. The method started with classifying the continuous (numeric) factors into five classes in ArcGIS 10.4. The FR score can be obtained by measuring the ratio between the number of pixel fire incidents with the number of pixel areas in each class. A particular class of the related variable has a high probability of wildfire occurrence if the class acquired an FR score larger than 1 and will have a significant effect on wildfire modeling. The FR score was used to create wildfire susceptibility models using the hybrid machine learning algorithms SVR-GWO and SVR-PSO, and SVR alone.

*2.6. Support Vector Regression (SVR)*

SVR is a derivative of the SVM method for a regression problem. Using a few samples, SVR finds a solution to nonlinear issues with high dimensionality based on structural risk minimization [65]. SVR performs faster convergence compared with other methods and finds a solution more efficiently in multidimensional estimation issues due to its ability to identify the relationship among input and output data [66]. For a set of wildfire data $\{x_i,\ y_i\}_i^n$, where *n* is the number of data points, $y_i$ are output values, and $x_i$ are input data. The relationship between input and output can be calculated based on a nonlinear function [*f*(*x*)] as follows:

$$y = f(x_i) = w^T \varphi(x_i) + b \tag{2}$$

where $w^T$ denotes the transpose value of the weight factor and *b* represents the bias vectors. $\varphi(x)$, as a nonlinear function, is utilized to map $x_i$. The controllable coefficients *w* and *b* are computed using Equations (4) and (5), as follows:

$$Minimize: \left[ \frac{1}{2}||w||^2 + C \sum_{i=1}^{n} \xi_i + \xi_i^* \right] \tag{3}$$

$$With\ the\ constrain: \begin{cases} y_i - (w^T \varphi(x_i) + b_i) \leq \varepsilon + \xi_i \\ (w^T \varphi(x_i) + b_i) - y_i \leq \varepsilon + \xi_i^* \\ \xi_i, \xi_i^* \geq 0 \quad i = 1, 2, \ldots, n \end{cases} \tag{4}$$

where *C* is a penalty factor or the trade-off the generalization capability and training error, $\xi_i$ and $\xi_i$* represent loose variables or distance between boundary and targets, and $\varepsilon$ is an insensitive loss function [65]. Lastly, the Lagrange equation can be used to solve the optimization issue, and the SVR function is expressed as:

$$f(x) = \sum_{i=1}^{n} (\alpha_i - \alpha_i^*) k(x, x_i) + b \tag{5}$$

where $\alpha_i$ and $\alpha_i$* are Lagrange multipliers, and $k(x,x_i) = \langle \varphi(x), \varphi(x_i) \rangle$ is called the kernel function. Accurately identifying the optimal value for the kernel function, $\varepsilon$ and *C* hyperparameters in the SVR are important to accomplishing the maximum accuracy of the model. Therefore, we used metaheuristic algorithms, including metaheuristic optimization algorithms including GWO and PSO, to tune and optimize the SVR hyperparameters.

*2.7. Metaheuristic Optimization Approaches*

2.7.1. Grey Wolf Optimization (GWO)

A GWO algorithm imitating the authority ranking and hunting operations of grey wolves (*Canis lupus*) was developed [67]. It is designed to identify the best solution for optimization problems. The three phases of hunting are start with searching for prey by defining the problems mathematically and identifying the basic parameters. The alpha ($\alpha$) is regarded as the highest- and best-fitted solution, and $\beta$, $\delta$, and $\omega$ represent the second, third and fourth best solution. The second step is the initialization of the pack randomly surrounding the entire space. The last step is attacking the prey when the prey is surrounded by the pack. The hunting operation is finished when an attack occurs and reaches scores between $-1$ and $1$ [34].

2.7.2. Particle Swarm Optimization (PSO)

PSO is a method that duplicates the intelligence of a swarm of insects, birds, or fish [68]. This method utilizes population fitness data to discover the best solution to a given issue. Each insect is considered a particle and is represented with a velocity and position vector. Every particle has its own intelligence and searches around in dimensional space to identify the best solution. After every iteration, each particle arranges its location by discovering the best location that it has ever visited and having the optimal proximity to its neighbor. For N-dimensional optimization problems, two vectors are considered for each particle, the *i*-th particle location (($\vec{X_i}$) = {$X_{i1}$, $X_{i2}$, ... , $X_{iN}$}), and the particle velocity (($\vec{V_i}$) = {$V_{i1}$, $V_{i2}$, ... , $V_{iN}$}), respectively. The location and velocity vectors are updated in each iteration as follows [69]:

$$\vec{V_i}(t+1) = w\vec{V_i}(t) + c_1 r_1 \times \left[ \vec{X_{Pi}} - \vec{X_i}(t) \right] + c_2 r_2 \times \left[ \vec{X_{Gi}} - \vec{X_i}(t) \right] \qquad (6)$$

$$\vec{X_i}(t+1) = \vec{X_i}(t) + \vec{V_i}(t+1) \qquad (7)$$

where $t$ is the iteration number, $w$ represents the inertial weight, $Pi$ refers to the best position of particle *i*-th, and $Gi$ represents the fittest position of the entire swarm. $c_1$ is the cognitive acceleration constant, while $c_2$ is the learning factor or social acceleration coefficient. $r_1$ and $r_2$ are two separate random coefficients ranging from 0 to 1 that are used to diversify the population.

*2.8. Performance Evaluation*

The evaluation step is a mandatory phase in analyzing the model prediction accuracy and performance to aid the scientific reliability of this study [70]. This study employed the root average squared error (RMSE) method as a cost function and as an evaluation metric for optimizing SVR hyperparameters, as follows:

$$RMSE = \sqrt{\frac{1}{n} \sum_{i=1}^{n} (p_i - o_i)^2} \qquad (8)$$

where $n$ indicates the number of samples and $p$ and $o$ represent the predicted and actual values of the wildfire inventory, respectively. Metaheuristic approaches were utilized to find the lowest RMSE scores through the optimization of SVR model hyperparameters.

This study also utilized the area under the receiver operating characteristic (ROC) curve (AUC) that has been employed to evaluate and validate global model assessment in machine learning and modeling research [71,72]. The AUC acts as an accurate indicator that reflects the performance, compression, and evaluation of model predictions. This calculation result ranges between 0.5 and 1, with scores close to 1 implying near-perfect performance and scores close to 0.5 indicating very weak predictive ability. We employed this approach using the testing dataset from wildfire inventory data, which was not utilized

in the training phase. The higher the AUC score and the closer it is to 1, the higher the performance of the wildfire prediction model.

### 3. Results

*3.1. Correlation between Wildfire and Driving Factors*

According to the Pearson correlation coefficient scores in Figure 7, the highest correlation score was calculated between forest type and land use (0.69). The diagonal unit in dark brown is the correlation between each factor and itself. Consequently, their score is equal to 1 [73]. The figure also shows that all of the correlation scores between every driving factor were outside of the critical values. This finding implied that there is no necessity to remove any driving factors and that all factors used in this study will not become a source of error or interference in the modeling phase.

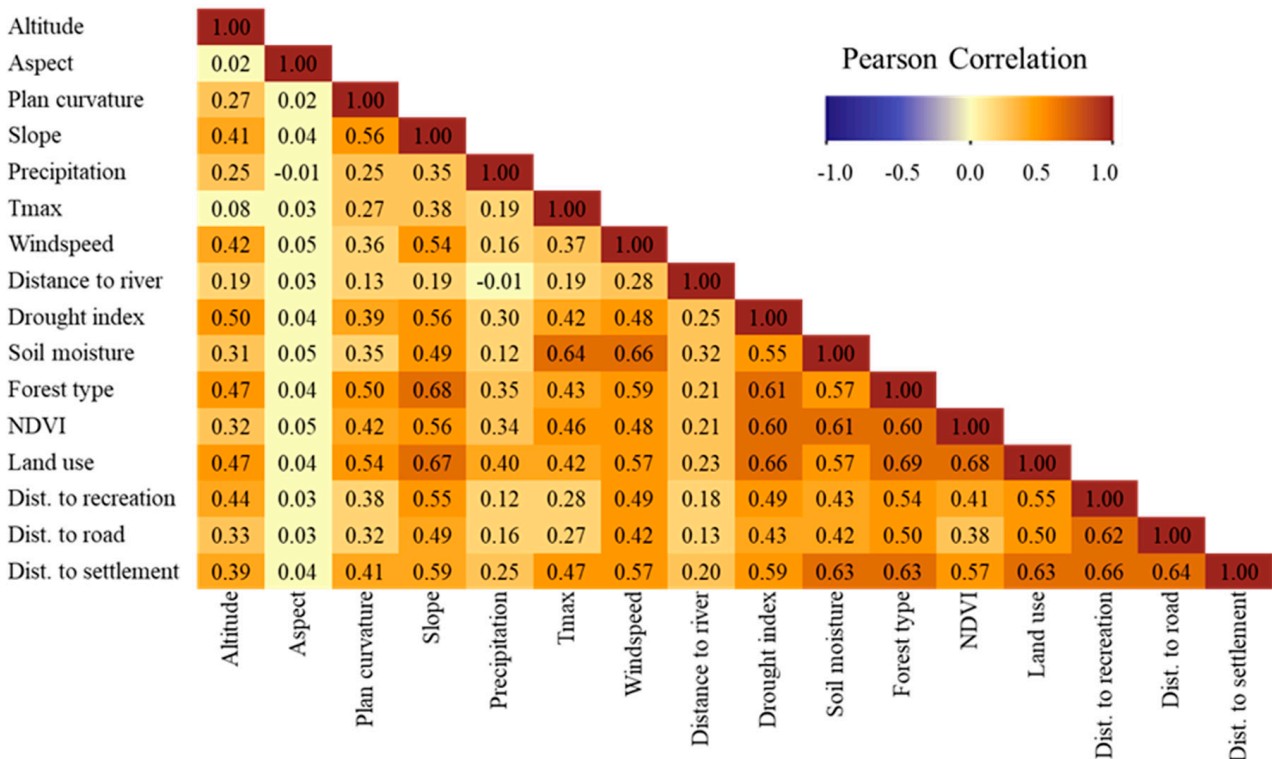

**Figure 7.** Pearson correlation between wildfire-driving factors.

Table 3 shows the results of a multicollinearity approach that was conducted to assess the correlation between wildfire occurrence with wildfire-driving factors. All 16 wildfire-driving factors had TOL results higher than 0.1 and VIF results lower than 10. The ranges of TOL and VIF scores were within the critical threshold. The highest VIF score was 3.86 for land use, and the lowest TOL score was 0.26 for land use. The range of TOL was between 0.26 to 0.99. The results suggested that no multicollinearities were monitored between the driving factors, thus avoiding the risk of decreasing model accuracy.

According to the assessment results of the IGR techniques, all 16 driving factors examined in this study had an impact on the wildfire occurrence and spread, and had predictive capabilities for wildfire modeling (IGR > 0), as shown in Table 3. Forest type factor was the highest IGR score (IGR = 0.85); thus, forest type was the most effective driving factor, followed by land use and slope degree with IGR scores of 0.78 and 054, respectively. These results were in agreement with the finding of prior studies that forest type, land use, and slope degree had the highest predictive power for wildfire susceptibility mapping [17,22,74]. Furthermore, this aspect was observed to have the lowest IGR score of 0.01. Considering that all Pearson correlation analysis, multicollinearity, and IGR scores

were within the safe threshold, all of the driving factors could be utilized for producing wildfire susceptibility modeling in this study.

**Table 3.** TOL, VIF, and IGR scores of wildfire-driving factors.

| Factor | Multicollinearity Scores | | IGR |
| --- | --- | --- | --- |
| | TOL | VIF | |
| Altitude | 0.61 | 1.65 | 0.24 |
| Aspect | 0.99 | 1.00 | 0.01 |
| Plan curvature | 0.64 | 1.55 | 0.25 |
| Slope | 0.37 | 2.72 | 0.54 |
| Precipitation | 0.74 | 1.35 | 0.17 |
| Tmax | 0.53 | 1.90 | 0.19 |
| Windspeed | 0.45 | 2.21 | 0.32 |
| Dist. to rivers | 0.87 | 1.15 | 0.04 |
| Drought index | 0.44 | 2.27 | 0.40 |
| Soil moisture | 0.31 | 3.24 | 0.34 |
| Forest type | 0.31 | 3.22 | 0.85 |
| NDVI | 0.43 | 2.34 | 0.49 |
| Land use | 0.26 | 3.86 | 0.78 |
| Dist. to recreational | 0.40 | 2.53 | 0.35 |
| Dist. to roads | 0.52 | 1.92 | 0.27 |
| Dist. to human settlements | 0.30 | 3.28 | 0.52 |

The spatial association between the historical location of wildfire incidents and classes of each driving factor using the FR method are shown in Table 4. The areas of 154–367 and 367–590 m altitude had higher FR scores of 1.54 and 1.66, respectively, implying that lowlands were predominantly prone to wildfire incidents, and 64% of all wildfires happened in the study area. The slope aspect showed that wildfire locations were generally distributed in all classes and were occurred more in the northern, northeastern, southwestern, and southern portions of Sydney. The north-facing aspects receive extra sun radiation that intensifies the temperature of fuel and the low level of fuel moisture in the Southern Hemisphere, which leads to fires. The plan curvature with convex and concave shapes was correlated with a high incidence of wildfires. These results agree with other studies that indicated that the likelihood of fire incidents may be higher on concave slopes and lower on flat terrain [13,17]. The two slope classes, 9.54–17.18 and 17.18–81.15, had FR scores of 1.51 and 2.00, respectively, suggesting that a high degree of inclination caused wildfires to spread faster to steep areas. The further distance to roads indicated a higher possibility of fire incidents, with the class 1663–12,385 having the highest FR score of 2.14. The further the distance from the road, the more troublesome it is for firetrucks to reach and put out the wildfire. High NDVI classes, namely, 0.69–0.76 and 0.76–0.89, had FR scores of 2.30 and 1.50, respectively, indicating wildfire incidents in areas with high greenness and fuel availability. Great wildfires occurred in relatively high classes of maximum temperature, namely, 30.09–31.19 and 31.19–32.77, with FR scores of 1.60 and 1.22, respectively. The highest class of distance to rivers (1294–7708) had an FR score of 1.46. The higher distances to recreational locations, namely, 11,473–17,920 and 17,920–41,130, had FR scores of 1.25 and 1.95, respectively. In addition to the distance to settlements, the higher class scores, namely, 2589–5319 and 5319–21,784, had higher FR scores of 1.12 and 2.63, respectively. For land use and forest-type factors, conservation and natural environmental areas experienced the highest wildfire occurrence, with Casuarina, Melaleuca, and Eucalypt forest types serving as fuel for wildfires.

The opposite patterns were found for the drought index, with the lower classes exhibiting higher FR scores of 1.44 and 1.67 for classes of −5.75−−5.50 and −5.50−−5.29, respectively. Lower windspeed classes, namely 0.54–3.65, 3.65–4.22, and 4.22–4.78, had higher FR scores of 1.37, 1.11, and 1.05, respectively. Soil moisture classes of 4.2–7.1 and 7.1–10.2 had FR scores of 1.26 and 1.9, respectively. The wildfire incidents for the precipitation

factor tended to be high in most classes. Topographical- and anthropological-related factors might diminish the effects of soil moisture level, precipitation, and windspeed, since none of them showed any spatial trend that was strongly correlated with the appearance or absence of wildfire incidents.

**Table 4.** Frequency ratio score.

| Variable Name | Class | Total % | Event % | FR Score |
|---|---|---|---|---|
| Altitude (m) | 0–154 | 19.83 | 6.02 | 0.30 |
| | 154–367 | 20.42 | 31.38 | 1.54 |
| | 367–590 | 20.22 | 33.60 | 1.66 |
| | 590–792 | 20.12 | 16.98 | 0.84 |
| | 792–1356 | 19.41 | 12.02 | 0.62 |
| Aspect | Flat | 0.35 | 0.02 | 0.07 |
| | North | 12.51 | 13.73 | 1.10 |
| | Northeast | 12.78 | 13.34 | 1.04 |
| | East | 13.89 | 13.19 | 0.95 |
| | Southeast | 12.24 | 12.02 | 0.98 |
| | South | 10.69 | 11.39 | 1.07 |
| | Southwest | 11.24 | 11.36 | 1.01 |
| | West | 13.23 | 12.06 | 0.91 |
| | Northwest | 13.08 | 12.88 | 0.98 |
| Plan curvature | Concave | 18.66 | 27.22 | 1.46 |
| | Flat | 45.59 | 27.13 | 0.60 |
| | Convex | 35.75 | 45.65 | 1.28 |
| Slope (degree) | 0–2.22 | 17.71 | 4.05 | 0.23 |
| | 2.22–4.77 | 20.99 | 8.65 | 0.41 |
| | 4.77–9.54 | 20.87 | 16.91 | 0.81 |
| | 9.54–17.18 | 20.26 | 30.54 | 1.51 |
| | 17.18–81.15 | 19.96 | 39.85 | 2.00 |
| Precipitation (mm) | 48.19–62.76 | 19.67 | 7.84 | 0.40 |
| | 62.76–69.35 | 20.64 | 21.99 | 1.07 |
| | 69.35–75.94 | 19.82 | 20.13 | 1.02 |
| | 75.94–84.27 | 20.21 | 24.08 | 1.19 |
| | 84.27–136.64 | 19.67 | 26.00 | 1.32 |
| Tmax (°C) | 25.54–28.63 | 19.09 | 16.66 | 0.87 |
| | 28.63–30.09 | 21.06 | 17.50 | 0.83 |
| | 30.09–31.19 | 22.02 | 35.12 | 1.60 |
| | 31.19–32.77 | 19.45 | 23.70 | 1.22 |
| | 32.77–35.90 | 18.38 | 7.01 | 0.38 |
| Windspeed (m/s) | 0.54–3.65 | 20.00 | 27.43 | 1.37 |
| | 3.65–4.22 | 20.00 | 22.13 | 1.11 |
| | 4.22–4.78 | 20.00 | 20.90 | 1.05 |
| | 4.78–5.45 | 20.00 | 16.83 | 0.84 |
| | 5.45–12.71 | 20.00 | 12.70 | 0.64 |
| Distance to rivers (m) | 30–180 | 22.80 | 21.77 | 0.95 |
| | 180–421 | 21.99 | 19.15 | 0.87 |
| | 421–752 | 19.03 | 15.58 | 0.82 |
| | 752–1294 | 18.91 | 18.22 | 0.96 |
| | 1294–7708 | 17.27 | 25.28 | 1.46 |
| Drought index | −6.75–−5.75 | 19.10 | 12.20 | 0.64 |
| | −5.75–−5.50 | 20.57 | 29.59 | 1.44 |
| | −5.50–−5.29 | 28.36 | 47.30 | 1.67 |
| | −5.29–−5.09 | 20.78 | 8.62 | 0.41 |
| | −5.09–−4.30 | 11.19 | 2.31 | 0.21 |
| Soil moisture (mm) | 1.85–4.20 | 20.21 | 7.41 | 0.37 |
| | 4.20–7.10 | 20.36 | 25.67 | 1.26 |
| | 7.10–10.20 | 21.96 | 41.81 | 1.90 |
| | 10.20–15.70 | 19.49 | 13.49 | 0.69 |
| | 15.70–69.40 | 17.98 | 11.62 | 0.65 |

**Table 4.** *Cont.*

| Variable Name | Class | Total % | Event % | FR Score |
|---|---|---|---|---|
| Forest type | Acacia | 0.04 | 0.01 | 0.32 |
| | Callitris | 0.19 | 0.05 | 0.25 |
| | Casuarina | 1.91 | 2.25 | 1.18 |
| | Eucalypt | 52.29 | 93.08 | 1.78 |
| | Mangrove | 0.02 | 0.00 | 0.00 |
| | Melaleuca | 0.07 | 0.11 | 1.67 |
| | Mixed species | 0.00 | 0.00 | 0.00 |
| | Non-forest | 38.33 | 1.01 | 0.03 |
| | Unallocated type | 0.17 | 0.13 | 0.78 |
| | Other native forests | 4.42 | 2.28 | 0.52 |
| | Rainforest | 0.93 | 0.41 | 0.45 |
| | Softwood plantation | 1.59 | 0.66 | 0.41 |
| | Hardwood plantation | 0.03 | 0.00 | 0.00 |
| NDVI | 0–0.55 | 20.01 | 0.27 | 0.01 |
| | 0.55–0.62 | 20.00 | 5.42 | 0.27 |
| | 0.62–0.69 | 20.00 | 18.38 | 0.92 |
| | 0.69–0.76 | 20.00 | 45.95 | 2.30 |
| | 0.76–0.89 | 19.99 | 29.98 | 1.50 |
| Land use | Nature conservation | 45.31 | 91.71 | 2.02 |
| | Grazing native forest | 24.85 | 6.33 | 0.25 |
| | Dryland agriculture | 17.02 | 0.44 | 0.03 |
| | Irrigated agriculture | 0.64 | 0.00 | 0.00 |
| | Intensive uses | 10.37 | 1.24 | 0.12 |
| | Open water | 1.81 | 0.28 | 0.15 |
| Dist. to recreational areas (m) | 30–2447 | 19.83 | 8.31 | 0.42 |
| | 2447–6638 | 20.72 | 14.44 | 0.70 |
| | 6638–11,473 | 20.63 | 15.47 | 0.75 |
| | 11,473–17,920 | 19.78 | 24.67 | 1.25 |
| | 17,920–41,130 | 19.05 | 37.11 | 1.95 |
| Dist. to roads (m) | 30–150 | 25.22 | 14.35 | 0.57 |
| | 150–402 | 18.99 | 14.09 | 0.74 |
| | 402–823 | 18.57 | 14.92 | 0.80 |
| | 823–1663 | 18.55 | 16.72 | 0.90 |
| | 1663–12,385 | 18.67 | 39.91 | 2.14 |
| Dist. to human settlements (m) | 30–456 | 18.92 | 4.49 | 0.24 |
| | 456–1309 | 22.00 | 9.68 | 0.44 |
| | 1309–2589 | 20.15 | 13.36 | 0.66 |
| | 2589–5319 | 19.85 | 22.28 | 1.12 |
| | 5319–21,784 | 19.08 | 50.19 | 2.63 |

*3.2. Susceptibility Map*

Figure 8 shows the results of wildfire susceptibility maps created employing the SVR alone and the hybrid metaheuristic optimized algorithms SVR-GWO and SVR-PSO. Wildfire susceptibility indices were generated for all pixels in the study area, where each pixel was assigned a unique susceptibility index. For visual inspection of wildfire susceptibility prediction, the quantile classification method was applied to categorize the pixel value with adjacent indexes into the same class and avoid the effect of subjective equal-interval classification [75,76]. Based on the modeling results and quantile method, each map was split into five predicted classes of very high, high, moderate, low, and very low wildfire susceptibility classes. All generated maps had almost identical outcomes in terms of the spatial extent of the very low and low susceptibility classes. Visually, the spatial pattern and distribution of wildfire susceptibility in the study area were strongly influenced by the slope (Figure 3d) and forest type (Figure 5d). Notable differences can be seen in Figure 8a, where middle areas were predicted as having moderate to high susceptibility. In Figure 8b,c, the areas were indicated as high to very high susceptibility to wildfire occurrence.

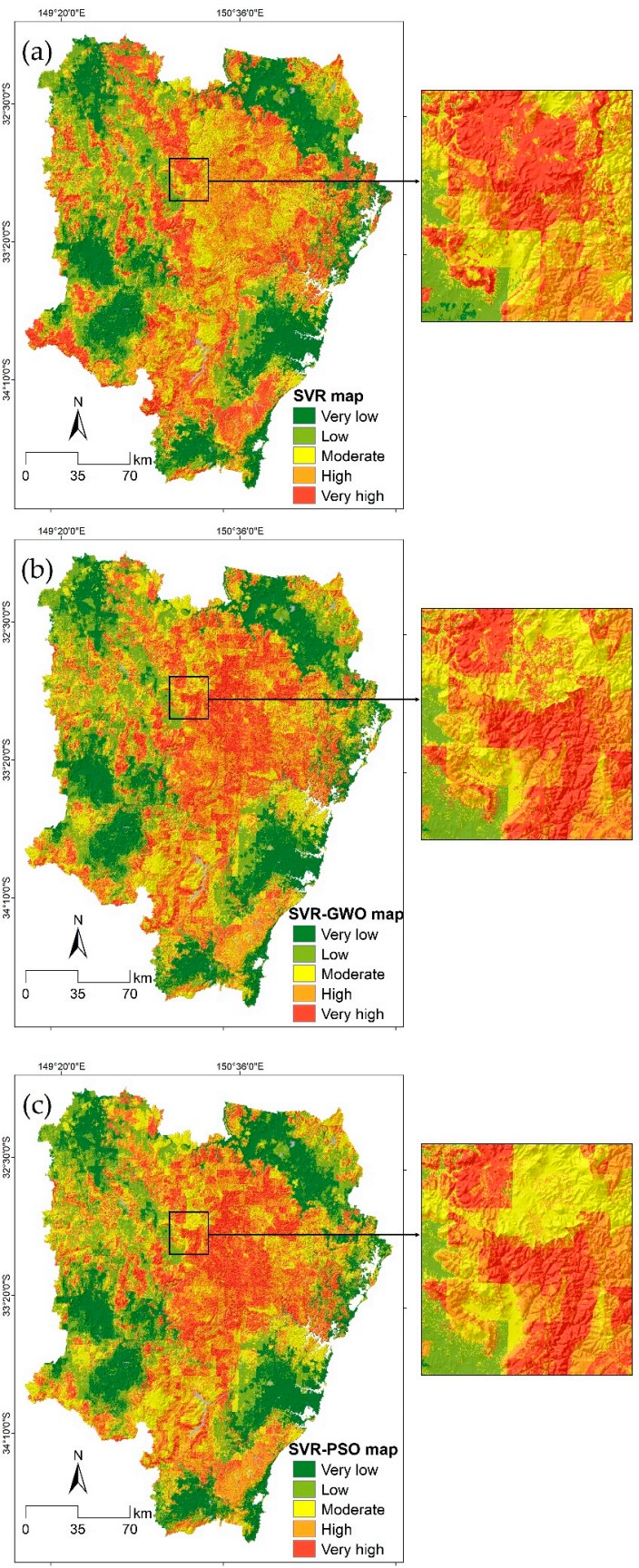

**Figure 8.** Wildfire susceptibility maps generated using the 2011–2018 wildfire data and (**a**) SVR, (**b**) SVR-GWO, and (**c**) SVR-PSO models.

Figure 9 shows the percentage of pixel distribution in every wildfire susceptibility map examined. The locations of very low and low class are similar in all maps and areas prone to wildfire incidents, particularly in the middle part of the study area and expanding to the south part, meaning that Sydney is surrounded by wildfire hotspot regions. Generally, approximately 40% of the study area has a low to very low wildfire susceptibility and is located in areas with low degrees of slope and low altitude; areas with moderate wildfire susceptibility were slightly different in the three models, and around 40% have a high and very high probability of wildfire incident because they are located at a high degree of slope. The distribution of pixels of SVR alone and hybrid metaheuristic optimized SVR-GWO maps showed similar results compared with the SVR-PSO map. The SVR-PSO map exhibited the greatest percentage of the high wildfire susceptibility class and the smallest percentage of the very high wildfire susceptibility class.

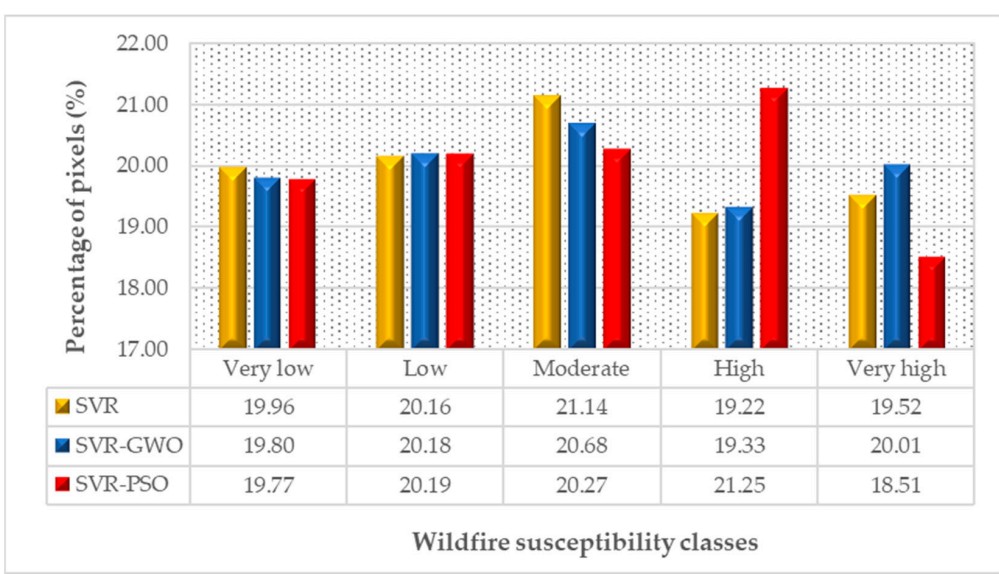

| | Very low | Low | Moderate | High | Very high |
|---|---|---|---|---|---|
| SVR | 19.96 | 20.16 | 21.14 | 19.22 | 19.52 |
| SVR-GWO | 19.80 | 20.18 | 20.68 | 19.33 | 20.01 |
| SVR-PSO | 19.77 | 20.19 | 20.27 | 21.25 | 18.51 |

**Figure 9.** Percentage of wildfire susceptibility classes in the SVR, SVR-GWO, and SVR-PSO maps.

### 3.3. Model Evaluation

The model performance was validated, evaluated, and compared using testing data set to assess the liability of the maps created by each proposed algorithm. The RMSE results showed that in the testing step applying the SVR model, the score of RMSE was 0.097. In SVR-PSO and SVR-GWO, the RMSE scores were 0.006 and 0.080, respectively. Furthermore, AUC score analysis was also used to validate the performance of the wildfire susceptibility models through model evaluation using 30% of the total dataset (testing dataset). Figure 10 shows the ROC curve analysis results that revealed the ROC of SVR (blue line), SVR-GWO (red line), and SVR-PSO (green line). The ROC curve graph shows the sensitivity (i.e., on the y-axis is true-positive) versus specificity (i.e., on the x-axis is false-positive). Figure 10 also shows that SVR has an AUC score of 0.837, SVR-GWO has an AUC score of 0.873, and SVR-PSO has an AUC score of 0.882. Given the higher scores of AUC and the lower scores of RMSE in the testing step, the accuracy and predictive capacity of the SVR-PSO model outperformed the SVR-GWO and standalone SVR models. This result is consistent with the findings from other studies [77].

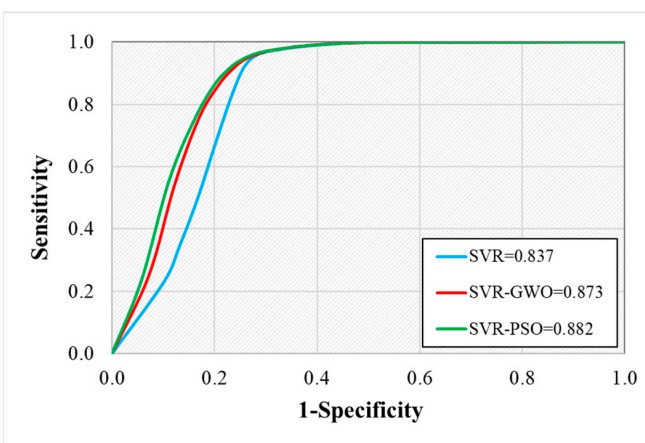

**Figure 10.** AUC scores and ROC curves of SVR, SVR-GWO, and SVR-PSO using a testing dataset.

## 4. Discussion

Susceptibility mapping with an appropriate assessment technique is a vital component for preventing and reducing wildfire damage. Wildfire inventory databases are essential for precise wildfire susceptibility construction. The remote sensing imagery from the Suomi NPP satellite using VIIRS is suitable for acquiring data inventories of wildfire incidents due to their availability, cost and time efficiency, independence from fieldwork, day and night daily data, and better coherent fire identification. Integrated with the fire perimeter from ABARES, a more accurate inventory dataset delineating the burned area after wildfires was acquired for training, testing, and evaluating the wildfire susceptibility models.

A wide variety of driving factors for wildfire prediction was identified and acquired according to prior studies and data availability. It was also important to conduct a three-step phase applying the multicollinearity test, the Pearson correlation analysis, and the IGR approach to eliminate inconsequent variables/factors, avoid possible bias, obtain a higher prediction quality, and identify important factors. The results showed that there was no notable multicollinearity and that all related factors utilized in this study possessed a significant influence on wildfire susceptibility in Sydney. The FR method was utilized, and the results revealed that the spatial relationship between each factor and wildfire occurrence was not spread randomly across Sydney. Regions with a high possibility of wildfires were related to slope, land use, and forest type. These results are in line with many studies [17]. Land use determines the type, amount, structure, and continuity of vegetation, with fuel characteristics and fuel continuity being variables that predispose the characteristics of wildfires, such as the probability, distribution, severity, and frequency, by supplying different fuel amounts and conditions for different times. Moreover, land uses such as urban areas, cultivation areas, wetlands, and open water in Sydney may stop the spread of wildfires due to the insufficient sustainable vegetation for them to spread. The forest type in Sydney has a significant effect on wildfires, particularly in eucalypt and areas exhibiting the highest FR score and wildfire occurrence, followed by the Melaleuca and Casuarina areas. Furthermore, the nature conservation areas with eucalypt forests had a high to very high possibility of wildfires in the created prediction maps. The fuel load is described as the forest type, such as the amount of leaf litter, small branches, and fallen bark that gathers in the landscape. Generally, the more abundant the fuel load is, the hotter and more intense the wildfire. Concentrated but loosely compacted fuel will burn quicker than highly dense or dispersed fuel sources. Smaller fuels, such as branches, twigs, and leaf litter, burn fast, especially if they are dry and untidy, and will burn faster in front of a fire. The eucalypt trees produce a natural oil that increases the combustion of fuel [78]. Wildfires spread faster when traveling uphill and decelerate when traveling downhill. The inclination of the slope plays a significant part in the speed of fire distribution. The rate of a fire front's spread doubles with every 10 degree increase in incline compared to a fire on a flat landscape. As the fire grows faster, it becomes more intense and more dangerous.

We presented SVR and hybrid machine learning algorithms, namely SVR-GWO and SVR-PSO, for modeling and identifying areas susceptible to wildfire in Sydney, Australia. All resulting models from this proposed framework exhibited good results with AUC scores greater than 0.8. The performance of SVR-PSO was the best, with AUC = 0.882, followed by SVR-GWO (AUC = 0.873) and standalone SVR (AUC = 0.837). The hybrid models achieved higher performance than the standalone models. The AUC values of the hybrid models were also better than those of a previous study in a similar area using SVM and SVMFR, which had AUC values of about 0.781 and 0.753. The hyperparameter tuning of the SVR algorithm utilizing a metaheuristic optimization algorithm affects the prediction and accuracy of the model. Thus, hybrid models are essential methods for enhancing the prediction capability of basic regression to decrease bias and prevent the issue of under-fitting and overfitting. Moreover, from the literature reviews, PSO algorithms generate an arbitrary solution and then find an accurate solution with an incremental optimum fitness attribute [79]. This type of approach has been used primarily for backpropagation (BP) genetic algorithms because of its easy installation efficiency, predictive accuracy, and fast response. It also indicates dominance in the resolution of complex practices, and was originally performed in a machine learning context. The best function of the PSO algorithm is to integrate various interconnected particles to achieve an optimal position. The same technique shows the highest position, velocity, and accuracy of each particle, which is determined by the basic concepts used to refine the problem. Particularly compared to other optimization techniques, the benefit of the PSO algorithm is that the PSO technique includes an important and fast search mechanism, is easy to use, and can identify the global optimal method that is closest to the best solution [79].

Figure 11 shows the SVR-PSO model, which performed better, compared with the 2019–2020 Black Summer fire season. One of the Black Summer fires was the Gospers Mountain Fire, which was caused by lightning and burned 479.513 ha before being put out after 107 days [80]. The RMSE score for the wildfire susceptibility map from the SVR-PSO model with the Black Summer fire season was 0.568, which was lower than that of the SVR-GWO (0.607) and standalone SVR models (0.608). The Black Summer fire season occurred in very high and high susceptible areas. Therefore, the hybrid model has better predictive capability in detecting future wildfires.

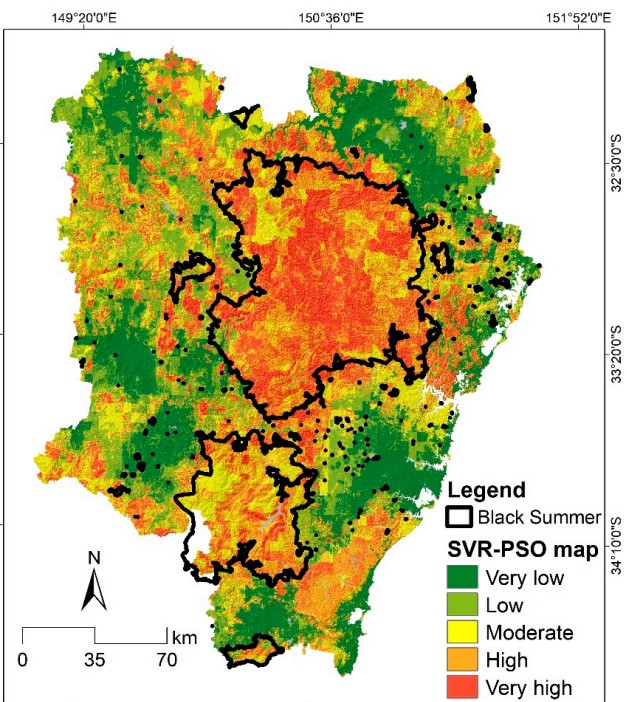

**Figure 11.** Comparison of the SVR-PSO map with the 2019–2020 Black Summer season.

## 5. Conclusions

The application of machine learning based on SVR models with metaheuristic optimization algorithms (GWO and PSO) integrated with VIIRS-Suomi data has been investigated for the spatial prediction of wildfire susceptibility for the first time in Sydney, Australia. The models were developed by establishing the correlation between 16 driving factors and 8 years of fire history from Suomi VIIRS data. Three model architectures appropriate for the prediction of wildfire susceptibility were proposed, and the SVR hyperparameters were optimized to enhance the prediction accuracy. Finally, the performances of the proposed models were compared using ROC curves and AUCs. Hybrid models with metaheuristic optimization algorithms will have important practical applications for wildfire mitigation programs in any area. The wildfire susceptibility results quantitively reveal which areas are susceptible to wildfire occurrence and provide information on wildfire status for regional management in Sydney. In addition, the spatial correlation analysis provides guidance for the determination of the strong factors influencing wildfires in the study area, such as land use, forest type, and slope degree, while the factors of aspect and distance to river have relatively small impacts on the models. This study has limitations. The precipitation, maximum temperature, drought index, and soil moisture datasets were collected from Terra Climate satellites with poor resolution; hence, the actual climate conditions and changes are not represented well, especially in regional level studies, causing uncertainty in making maps of wildfire susceptibility. Since this study makes some assumptions, such as the effect of the slope degree on the speed of fire spread, fuel conditions, and low resolution data, future research should provide field data from the study area on factors related to wildfires to provide more accurate data. Based on the proposed SVR model that optimized the hyperparameters by metaheuristic algorithms, we acquired higher prediction accuracy results of wildfire susceptibility, which serves as a reference for subsequent regional wildfire hazard assessment. The hybrid model based on machine learning with metaheuristic optimization algorithms has the potential to be applied to different natural hazards to create susceptibility maps.

**Author Contributions:** Conceptualization, C.-W.L. and A.S.N.; methodology, C.-W.L. and A.S.N.; software, C.-W.L. and A.S.N.; validation, C.-W.L., Y.J.K. and A.S.N.; formal analysis, C.-W.L. and A.S.N.; investigation, C.-W.L., J.H.L. and A.S.N.; resources, C.-W.L., Y.J.K., J.H.L. and A.S.N.; data curation, C.-W.L., Y.J.K., J.H.L. and A.S.N.; writing—original draft preparation, A.S.N.; writing—review and editing, C.-W.L., Y.J.K., J.H.L. and A.S.N.; visualization, C.-W.L. and A.S.N.; supervision, C.-W.L., J.H.L. and Y.J.K.; project administration, J.H.L. and C.-W.L.; funding acquisition, J.H.L. and C.-W.L. All authors have read and agreed to the published version of the manuscript.

**Funding:** This research was supported by a grant from the Korea Polar Research Institute (KOPRI, PE22900) and the National Research Foundation of Korea (No. 2019R1A6A1A03033167).

**Data Availability Statement:** Data supporting the reported results can be found at https://firms.modaps.eosdis.nasa.gov/ for fire data from VIIRS-Suomi data, accessed on 17 October 2022; https://portal.opentopography.org/raster?opentopoID=OTSDEM.032021.4326.3 for Copernicus DEM data, accessed on 17 October 2022; https://globalwindatlas.info/en for windspeed data, accessed on 17 October 2022; https://datasets.seed.nsw.gov.au/dataset/fire-history-wildfires-and-prescribed-burns-1e8b6 for fire perimeter data, accessed on 17 October 2022; and https://data-fcnsw.opendata.arcgis.com/search?collection=Dataset for forest type data, accessed on 17 October 2022.

**Conflicts of Interest:** The authors declare that they have no conflict of interest. The funders had no role in the design of the study; in the collection, analyses, or interpretation of data; in the writing of the manuscript; or in the decision to publish the results.

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
