# Peer review of "Spatial Prediction of Wildfire Susceptibility Using Hybrid Machine Learning Models Based on Support Vector Regression in Sydney, Australia"

_remotesensing, doi:10.3390/rs15030760_

Round 1
Reviewer 1 Report
Manuscript ID: remotesensing-2057060
Title: Spatial Prediction of Wildfire Susceptibility using Hybrid Machine Learning Models Based on Support Vector Regression in Sydney, Australia
The authors have presented the application of machine learning based on SVR models with metaheuristic optimization algorithms (GWO and PSO) integrated with VIIRS-Suomi data for the spatial prediction of wildfire susceptibility. Obtained data will be useful for research, wildfire risk assessment, and fuel management operations. The authors have provided a nice background of the problem and introduced the objectives. Work is of high quality. However, I have some minor suggestions before the study can be accepted for publication. Please address them separately and highlight the changes in the manuscript.
Comments:
1. Can authors provide a comparison of their model with previous studies and highlight the accuracy of new, improved models?
2. Include the nomenclature section at the beginning.
3. Minor changes in English style may be required, especially few sentences are either too long or too short.
4. Since the study has some assumptions, I would recommend highlighting them all in the conclusion sections and maybe small discussion on them with future works.
5. Formatting of the references as per the journal style.
Author Response
Response Reviewer 1
Round 1
Title: Spatial Prediction of Wildfire Susceptibility using Hybrid Machine Learning Models Based on Support Vector Regression in Sydney, Australia
The authors have presented the application of machine learning based on SVR models with metaheuristic optimization algorithms (GWO and PSO) integrated with VIIRS-Suomi data for the spatial prediction of wildfire susceptibility. Obtained data will be useful for research, wildfire risk assessment, and fuel management operations. The authors have provided a nice background of the problem and introduced the objectives. Work is of high quality. However, I have some minor suggestions before the study can be accepted for publication. Please address them separately and highlight the changes in the manuscript.
Response: We appreciate your meticulous review and constructive comments provided by the reviewer and we have made a modification accordingly. Our responses are in red. Line numbers in red refer to line numbers in the revised “Tracked changes” manuscript. Also, the changes in the manuscript were highlighted in yellow.
Comments:
- Can authors provide a comparison of their model with previous studies and highlight the accuracy of new, improved models?
Response: Thank you for your suggestion, we have provided a comparison of our model with previous studies and highlight the accuracy of new improved models.
Line 526-528, “The AUC values of hybrid models were also better than that of a previous study in a similar area using SVM and SVMFR, which had AUC values of about 0.781 and 0.753.”
- Include the nomenclature section at the beginning.
Response: Thank you. As you suggest, we have included the nomenclature table in the introduction section.
Line 111-113, “As a matter of convenience, Table 1 represents the nomenclature of this paper.
Table 1. This table represents the nomenclature of this paper.
|
Nomenclature |
|||
|
MODIS |
moderate resolution imaging spectroradiometer |
H |
training dataset |
|
VIIRS |
visible infrared imaging radiometer suite |
RF |
related factors |
|
GIS |
geographic information system |
FR |
frequency ratio |
|
RS |
remote sensing |
n |
number of data points or samples |
|
AHP |
analytical hierarchy process |
yi |
output values |
|
ANN |
artificial neural network |
xi |
input data |
|
WOE |
weight of evidence |
the transpose value of weight factor |
|
|
SVR |
support vector regression |
b |
bias vectors |
|
GWO |
grey wolf optimization |
φ(x) |
nonlinear function |
|
PSO |
particle swarm optimization |
C |
penalty factor |
|
FIRMS |
fire information for resource management system |
ξi |
loose variables or distance between boundary |
|
RMSE |
root mean square error |
ξi* |
targets |
|
ROC |
the receiver operating characteristic |
ε |
insensitive loss function |
|
AUC |
area under ROC curve |
αi |
Lagrange multipliers |
|
NSW |
New South Wales |
k(x,xi) |
the kernel function |
|
Cfa |
humid subtropical climate |
particle location |
|
|
S-NPP |
Suomi-national polar-orbiting partnership |
particle velocity |
|
|
GPS |
global positioning system |
t |
iteration number |
|
NDVI |
normalized difference vegetation index |
w |
inertial weight |
|
DEM |
digital elevation model |
Pi |
the best position of particle i |
|
ABARES |
Australian bureau of agricultural and resource economics and sciences |
G |
the fittest position of the entire swarm |
|
PDSI |
Palmer drought severity index |
c1 |
cognitive acceleration constant |
|
ACLUMP |
Australian collaborative land use management program |
c2 |
social acceleration coefficient |
|
IGR |
information gain ratio |
r |
random coefficients range from 0 to 1 |
|
TOL |
tolerance |
p |
predicted value |
|
VIF |
variance inflation factor |
o |
actual value |
- Minor changes in English style may be required, especially few sentences are either too long or too short.
Response: Thank you for the comments, we have made modifications to some sentences for better understanding.
- Since the study has some assumptions, I would recommend highlighting them all in the conclusion sections and maybe small discussion on them with future works.
Response: Thank you, as you suggest we have added a sentence in conclusion section.
Line 574-576, “Since this study has some assumptions, future research should provide field data regarding the wildfire-related factors to provide more accurate data.”
- Formatting of the references as per the journal style.
Response: Thank you, we have corrected the references following the journal style.

Reviewer 2 Report
This paper examined a wildfire susceptibility mapping using various data for the land surface environment, which is a reasonable approach and can be considered publication after some modifications regarding additional explanations of the method used.
2.6.1. Grey Wolf Optimization (GWO) and 2.6.2. Particle Swarm Optimization (PSO): In addition to the textbook principles, please describe how you used the GWO and PSO in this study.
3.2. Susceptibility Map. This section needs a justification that the quantile classification is best for wildfire susceptibility.
Figure 8. The authors can also present the maps for the differences between A and B, B and C, and A and C.
Author Response
Response Reviewer 2
Round 2
This paper examined a wildfire susceptibility mapping using various data for the land surface environment, which is a reasonable approach and can be considered publication after some modifications regarding additional explanations of the method used.
Response: We appreciate your meticulous review and constructive comments provided by the reviewer and we have made a modification accordingly. Our responses are in red. Line numbers in red refer to line numbers in the revised “Tracked changes” manuscript. Also, the changes in manuscript were highlighted in yellow.
2.6.1. Grey Wolf Optimization (GWO) and 2.6.2. Particle Swarm Optimization (PSO): In addition to the textbook principles, please describe how you used the GWO and PSO in this study.
Response: Thank you for your constructive comments, GWO and PSO were used to automatically identify the optimal value for the kernel function, ε, and C, since they have considerable effect on forecasting accuracy in SVR model.
Line 313-316, “Accurately identifying the optimal value for the kernel function, ε, and C hyperparameter in the SVR are important to accomplishing the maximum accuracy of the model. Therefore, we used metaheuristic algorithms, including metaheuristic optimization algorithms including GWO and PSO to tune and optimize the SVR hyperparameters.”
3.2. Susceptibility Map. This section needs a justification that the quantile classification is best for wildfire susceptibility.
Response: Thank you, we already gave a justification that the quantile classification is best for wildfire susceptibility mapping.
Line 433-440, “Wildfire susceptibility indices were generated for all pixels in the study area, where each pixel was assigned a unique susceptibility index. For visual inspection of wildfire susceptibility prediction, the quantile classification method was applied to categorize the pixel value with adjacent indexes into the same class and avoid the effect of subjective equal-interval classification [73,74]. Based on the modeling results and quantile method, each map was split into five predicted classes of very high, high, moderate, low, and very low wildfire susceptibility classes.”
Figure 8. The authors can also present the maps for the differences between A and B, B and C, and A and C.
Response: Thank you, we have made modification in Figure 8 to present the differences between each map.
Line 447, “
Figure 8. Wildfire susceptibility maps generated using the 2011-2018 wildfire data and (a) SVR, (b) SVR-GWO, and (c) SVR-PSO models.”

Reviewer 3 Report
Overall, the paper is interesting in that it provides unique methodology for fire risk / susceptibility. Specifically, the use of the 2 optimization methods for a hybrid machine learning approach is interesting and not yet applied in this domain. Currently, the accuracy assessment of the paper is based on a single large combination of fire events from 1 season, while this is useful, it is not a proper validation of the susceptibility model because of several reasons: 1) needs a randomization component to accuracy assessment 2) need to cover other time periods. I suggest authors to use VIIRS/MODIS burned area or active fire products for evaluation of accuracy similar to the work in Sepideh Tavakkoli Piralilou et al. 2022
Other comments:
Paragraph near line 156: It was previously mentioned that fire points of only high confidence were selected. For the non-fire points, this paragraph mentions selection of random locations outside of these high confidence fire points. Regarding this, two questions: 1) Are you including low and normal confidence fire points in non-fire locations? If so, why/why not. 2) Are non-fire locations selected based on random samplings or stratified by phenomena such as land cover? if so why/why not.
Section 2.6 on optimization methods need to be described in context/application of wildfire prediction and not in animal behavior.
Line 274-275 needs citation for critical value for variable colinearity. Model does appear to have some colinearity especially some of the similar variables above 0.65
Please check the spelling, grammar, etc. as there are errors.
Author Response
Response Reviewer 3
Round 1
Overall, the paper is interesting in that it provides unique methodology for fire risk / susceptibility. Specifically, the use of the 2 optimization methods for a hybrid machine learning approach is interesting and not yet applied in this domain. Currently, the accuracy assessment of the paper is based on a single large combination of fire events from 1 season, while this is useful, it is not a proper validation of the susceptibility model because of several reasons: 1) needs a randomization component to accuracy assessment 2) need to cover other time periods. I suggest authors to use VIIRS/MODIS burned area or active fire products for evaluation of accuracy similar to the work in Sepideh Tavakkoli Piralilou et al. 2022
Response: We appreciate your meticulous review and constructive comments provided by the reviewer and we have made a modification accordingly. Our responses are in red. Line numbers in red refer to line numbers in the revised “Tracked changes” manuscript. Also, the changes in manuscript were highlighted in yellow. For the accuracy assessment of the paper, we used testing dataset which is 30% from the 2011-2018 fire data using random function. This evaluation process has been done by many previous studies for creating wildfire susceptibility models, published in high rank journals (Q1). The single large combination fire events from 2019-2020 Black Summer fire season were used to evaluate the predictive capability of the models. We will consider to use VIIRS/MODIS burned area or active products for evaluation accuracy for future study based on the reference.
Nur, A.S.; Kim, Y.J.; Lee, C.-W. Creation of Wildfire Susceptibility Maps in Plumas National Forest Using InSAR Coherence, Deep Learning, and Metaheuristic Optimization Approaches. Remote Sens. 2022, 14, 4416. https://doi.org/10.3390/rs14174416
Hong, H.; Jaafari, A.; Zenner, E.K. Predicting Spatial Patterns of Wildfire Susceptibility in the Huichang County, China: An Integrated Model to Analysis of Landscape Indicators. Ecol Indic 2019, 101, 878–891, doi:10.1016/j.ecolind.2019.01.056.
Al-Fugara A, Mabdeh AN, Ahmadlou M, Pourghasemi HR, Al-Adamat R, Pradhan B, Al-Shabeeb AR. Wildland Fire Susceptibility Mapping Using Support Vector Regression and Adaptive Neuro-Fuzzy Inference System-Based Whale Optimization Algorithm and Simulated Annealing. ISPRS International Journal of Geo-Information. 2021; 10(6):382. https://doi.org/10.3390/ijgi10060382.
Other comments:
Paragraph near line 156: It was previously mentioned that fire points of only high confidence were selected. For the non-fire points, this paragraph mentions selection of random locations outside of these high confidence fire points. Regarding this, two questions: 1) Are you including low and normal confidence fire points in non-fire locations? If so, why/why not. 2) Are non-fire locations selected based on random samplings or stratified by phenomena such as land cover? if so why/why not.
Response: Thank you for your comments, we did not include the low and normal confidence fire points in non-fire locations, because these points are possibly fire but with low certainty. As for non-fire location, we generated using Frequency Ratio method using all related factors with the low probability of fire location and filtered outside fire perimeter. These combined methods were considered to provide better, more certain, and accurate fire location information.
Line 152-154, “Only high confidence level data were used as true fire hotspots for the wildfire inventory database to provide more certain and accurate fire location information.”
Line 162-166, “Equal number of nonwildfire location data (16,462 points) were picked using random point function by identifying the region outside previous wildfire history and having very low possibility areas determined using frequency ratio approach. This approach was an effective strategy to help the interpretation of the area and provides a more precise wildfire inventory.”
Section 2.6 on optimization methods need to be described in context/application of wildfire prediction and not in animal behavior.
Response: Thank you for your comments, we have added the description in context/application of wildfire prediction.
Line 313-316, “Accurately identifying the optimal value for the kernel function, ε, and C hyperparameter in the SVR are important to accomplishing the maximum accuracy of the model. Therefore, we used metaheuristic algorithms, including metaheuristic optimization algorithms including GWO and PSO to tune and optimize the SVR hyperparameters.”
Line 274-275 needs citation for critical value for variable colinearity. Model does appear to have some colinearity especially some of the similar variables above 0.65
Response: Thank you. As you suggest, we have added references for the critical value for variable collinearity. According to several studies that the collinearity occurs when the value reaches 0.7. However, our collinearity score under 0.7. Therefore, we considered that no multicollinearity occurred between variables.
Line 276-279, “Scores equal to 0 indicate no correlation between the two factors. Prior studies have considered the scores – 0.7 and 0.7 to be the critical score, implying that scores above these critical scores may lead to a multicollinearity problem [17,60,61].
Please check the spelling, grammar, etc. as there are errors.
Response: Thank you, we have checked and corrected the spelling, grammar, and other errors.
